# Preparation and Evaluation of Mucus-Penetrating Inhalable Microparticles of Tiotropium Bromide Containing Sodium Glycocholate

**DOI:** 10.3390/pharmaceutics14071409

**Published:** 2022-07-05

**Authors:** Yong-Bin Kwon, Ji-Hyun Kang, Young-Jin Kim, Dong-Wook Kim, Sung-Hoon Lee, Chun-Woong Park

**Affiliations:** 1College of Pharmacy, Chungbuk National University, 194-21, Osongsangmyeong 1-ro, Heungdeok-gu, Cheongju 28160, Korea; yongbin6474@naver.com (Y.-B.K.); jhkanga@naver.com (J.-H.K.); 7777ytrewq@gmail.com (Y.-J.K.); 2College of Pharmacy, Wonkwang University, Iksan 54538, Korea; pharmengin@gmail.com; 3Department of Pharmaceutical Engineering, Cheongju University, Cheongju 28503, Korea; sunghoon@cju.ac.kr

**Keywords:** tiotropium, sodium glycocholate, permeation enhancer, aerodynamic properties, Calu-3 cell, asthma-induced rat model

## Abstract

This study aimed to prepare mucus-penetrating inhalable microparticles for dry powder inhalers and to evaluate their applicability in an asthma-induced rat model. Microparticles were prepared from water solutions containing tiotropium bromide, L-leucine, and sodium glycocholate (NaGc) as permeation enhancers using the spray drying method. Four formulations (SDL1, SDL2, SDL3, and SDL4) were used, depending on the various NaGc concentrations. Tiotropium microparticles were characterized by standard methods. Additionally, an asthma-induced rat model was used to confirm the effects of the formulations on lung function. Tiotropium microparticles with NaGc resulted in formulations with a more corrugated morphology and smaller particle size distribution than those without NaGc. SDL 1 had a rough surface with irregular morphology, and SDL 2, 3, and 4 had a corrugated morphology. All SDL formulations had an aerodynamic size of <3 µm. The microparticles with a corrugated morphology aerosolized better than SDL1 microparticles. The apparent permeability coefficient (P_app_) values of SDL3 and SDL4 were significantly higher than those for raw tiotropium. In an in vivo study using an asthma-induced rat model, the specific airway resistance (S_raw_), airway wall thickness, and mean alveolus size recovered to those of the negative control group in the SDL4 formulation.

## 1. Introduction

To date, many methods have been developed for pulmonary drug delivery to treat asthma and other pulmonary disorders. Pulmonary delivery systems can be classified into three types: dry powder inhalers (DPIs), pressurized metered-dose inhalers (pMDIs), and nebulizers. The efficiency of DPIs is usually determined by the aerosol performance of the particles, and the aerosol performance is influenced by various factors, such as particle size distribution, surface morphology, particle shape, interparticulate forces, crystallinity, and water content. The aerodynamic diameter of the particles is 1–5 µm for deep lung delivery, which is considered respirable [1,2]. As reported previously, microparticles can be prepared according to several physical and chemical methods [3]. The spray drying technique has been successfully introduced to produce microparticles for lung delivery systems [4,5,6,7,8,9]. In this method, dry powder can be obtained using a rapid and single-step process from the drug solution. In addition, microparticles with the desired size and morphology can be obtained by adjusting the spray drying parameters and feeding solution conditions (i.e., feeding rate, solvent type, and concentration) [10,11].

In this study, we prepared absorption-enhanced tiotropium bromide (TB) for DPIs to deliver the drug to the bronchioles and alveoli. To deliver TB to the peripheral side of the lung, the drug particle size must be micronized to a respirable particle size (i.e., 1–5 µm aerodynamic diameter) [12,13].

TB is a long-acting antimuscarinic bronchodilator that is used in chronic obstructive pulmonary disease (COPD) and asthma. It is to be administered once a day and is known to improve lung function, quality of life, and exacerbation frequency [14]. Absorption-enhanced TB microparticles can penetrate the lung mucus barrier and biological epithelium. Mucus in COPD and asthma patients has higher viscosity and elasticity than that in individuals without lung disease [15]. Thus, the particle-based drugs are more firmly trapped in the mucosal layer [16]. The stronger interactions of mucus–drug may affect the fate and distribution on the targeted site of pulmonary delivered drug particles, subsequently reducing the therapeutic efficacy [17].

In this study, we used sodium glycocholate as a permeability enhancement. It has been shown to increase the efficacy of active compounds, peptides, and proteins with low barrier permeability [18]. Bile salts play an important role as physiological surfactants and have been used to enhance the permeation of drugs through rectal, nasal, pulmonary, and ocular [19,20,21] routes. For pulmonary drug delivery applications, glycocholate, taurocholate, glycodeoxycholate, and taurodeoxycholate salts [22] have been tested as absorption enhancers. Among the bile salts used as absorption enhancers, sodium glycocholate (NaGc) was employed because of its efficacy and safety at the investigated concentrations [23]. The mechanism of absorption enhancement by bile salts has not been fully elucidated, and some studies have suggested that bile salts reduce the transepithelial electrical resistance (TEER) values and viscoelasticity of mucus, and consequently increase membrane permeability [24,25]. 

To assess the absorption-enhanced effect of inhalation, an in vitro Calu-3 cell monolayer was used to confirm the absorption-enhanced effects of TB across respiratory epithelial cells. Air–liquid interface cell models were used to track the fate of drugs and paracellular or transcellular processes across the pulmonary epithelia after direct deposition into the lung [26,27,28]. The Calu-3 monolayer cell model was used in combination with a modified impactor, such as an Anderson cascade impactor (ACI), multistage liquid impinger (MSLI), and twin stage impinger (TSI), to deposit aerosolized microparticles and evaluate drug absorption processes across epithelial cells.

In addition, we confirmed the effects of TB microparticles containing NaGc as a permeation enhancer on lung function, airway hyper-responsiveness, and histological changes in an ovalbumin (OVA)-induced rat model. The rats were sensitized and challenged with OVA at regular intervals. 

## 2. Materials and Methods

### 2.1. Materials

Micronized tiotropium bromide monohydrate was purchased from Vamsi Labs (Maharashtra, India). Calu-3 cells were purchased from American Type Cell Culture Collection (ATCC, Rockville, MD, USA). Dulbecco’s modified Eagle’s medium (DMEM, without phenol red and L-glutamine), penicillin–streptomycin (10,000 U/mL), fetal bovine serum (FBS), phosphate-buffered saline (PBS), Hanks balanced salt solution (HBSS), and 0.25% trypsin-EDTA (1×) were purchased from Gibco^®^ (New York, NY, USA). Alcian blue, Entellan^™^ New Rapid Mounting Medium, albumin from chicken egg white (Ovalbumin), methacholine chloride, and L-glutamine were obtained from Sigma-Aldrich (Sydney, Australia). Transwell^®^ cell culture inserts (0.33 cm^2^ polyester membrane, 0.4 µm pore size) and T-75 cell culture flasks were purchased from Corning Coastar^®^ (Lowell, MA, USA). The experimental animals (8-week-old male Sprague-Dawley rats) were purchased from Samtaco Bio Co. (Osan, Korea). High-performance liquid chromatography (HPLC)-grade ethanol and acetonitrile were used (Honeywell Burdick & Jackson^®^, Muskegon, MI, USA), and all other reagents were of analytical or HPLC grade.

### 2.2. Preparation of Spray-Dried Tiotropium Microparticles and Physical Mixture

Spray-dried TB with L-leucine and NaGc (SDL) was prepared using a spray dryer (EYELA SD-1000; Rikakikai Co., Ltd., Tokyo, Japan). As shown in Table 1, the TB, NaGc, and L-leucine were dissolved in distilled water (1.5% *w*/*w*) and spray dried with the following parameters: inlet temperature of 110 °C, nozzle (two-way nozzle spray type) size of 0.4 mm, feeding rate of 1.5 mL/min, atomization air pressure 250 kPa, drying air flow rate of 0.4 m^3^/min. For PM preparation, each component was accurately weighed in the same ratio as SDL 4, and vortexed for 5 min. To confirm the content uniformity, we transferred approximately 10 mg of SDL to a 50 mL volumetric flask, diluted it with mobile phase (MP) to volume, and sonicated it for 10 min. The content of tiotropium was analyzed using validated HPLC methods. Briefly, the assay was achieved under optimized chromatographic conditions on a C_18_ reversed-phase column (250 × 4.6 mm, 5 µm) eluted with MP (buffer pH 3.0: acetonitrile, 65:35, *v*/*v*) at a flow rate 1.5 mL/min at ambient temperature with UV detection at 230 nm using a 20 µL injection volume. The buffer was prepared by dissolving 1.38 g of sodium dihydrogen phosphate and 1.22 g of decane sulphonic acid in 1000 mL of distilled water and adjusting to pH 3.0 with orthophosphoric acid.

### 2.3. Physical and Chemical Characterization

#### 2.3.1. Scanning Electron Microscopy (SEM)

The spray-dried TB microparticles were visually imaged using SEM at 3 keV (Zeiss Ultra Plus, Carl Zeiss Pty Ltd., Sydney, Australia). Prior to imaging, the samples were dispersed onto carbon tapes and coated with platinum using a Hummer VI sputtering device, reaching 200 Å coating thickness. Magnifications of 3000×, 10,000×, and 25,000× were used.

#### 2.3.2. Particle Size Distribution via Laser Diffraction (PSD)

The particle size and particle size distribution of the TB and spray-dried TB microparticles were determined by laser diffraction particle sizing using a Mastersizer 2000 (Malvern Instruments, Worcestershire, UK). TB microparticle analysis was carried out by the wet dispersion method after dispersing the samples in a non-swelling solvent, isopropyl alcohol.

#### 2.3.3. Powder X-ray Powder Diffraction (PXRD)

The PXRD patterns of the spray-dried TB microparticles, physical mixture (PM), L-leucine, and sodium glycocholate were analyzed using an X-ray diffractometer (XDS 2000, SCINTAG, Cupertino, CA, USA) at a wavelength of 1.54. The 2*θ* scans were conducted between 5° and 50° with a step size of 0.009°/2*θ* at ambient temperature with a Cu radiation source (40 kV, 40 mA).

#### 2.3.4. Differential Scanning Calorimetry (DSC)

The thermal properties of the spray-dried TB microparticles, physical mixture (PM), L-leucine, and sodium glycocholate were analyzed using DSC (DSC 2910, TA Instruments, New Castle, DE, USA). Each sample was accurately weighed, placed in DSC aluminum pin hole pans, and analyzed at a heating rate of 10 °C/min over a temperature range of 20 °C to 350 °C.

#### 2.3.5. Fourier Transform Infrared (FT-IR) Spectroscopy

Infrared spectroscopy was performed using FT-IR spectroscopy (IFS 66v/S, Bruker Optics, Ettlingen, Germany) using the potassium bromide technique and a deuterated triglycine sulfate (DTGS) detector. Spectra were collected over the range 500–3500 cm^−1^.

### 2.4. In Vitro Aerosol Dispersion Performance by Andersen Cascade Impactor (ACI)

The in vitro aerosol dispersion performance of the SDL microparticles was determined using an eight-stage non-viable ACI without a pre-separator (TE-20-800, TISCH Environmental, Cleves, OH, USA) and RS01 (Plastiape, Osnago, Italy) as the DPI device, in accordance with the USP Chapter ‘601’ specification on aerosols. Each formulation was aerosolized by drawing air using a vacuum pump at a controlled flow rate of 60 L/min for 4 s (TISCH Environmental, Cleves, OH, USA). The 4 kpa pressure drop over the inhaler was checked using a critical flow controller (COPLEY Scientific, Colwick, UK). A hydroxypropyl methylcellulose hard capsule (size 3) was loaded with 10 mg of sample and placed in RS01, tightly inserted into the mouthpiece of the induction port. The collection plates of all stages in the ACI were pre-coated with silicone oil to prevent bounce and re-entrainment. The number of particles remaining in the capsule and deposited on each collection plate at all stages in ACI was measured. The HPMC capsule and collection plate of each stage were transferred to petri dishes (90 × 15 mm) and tiotropium was completely dissolved with 10 mL MP solution prior to analysis using validated HPLC methods. The aerodynamic cut-off diameters of each stage were stated as 8.6 µm, 6.5 µm, 4.4 µm, 3.3 µm, 2.0 µm, 1.1 µm, 0.54 µm, and 0.25 µm for stages −1 to 6, respectively. The emitted dose (ED) indicates the amount of dry powder released from the capsule. The fine particle fraction (FPF) represents the mass percentage of dry powders to reach the respirable region with aerodynamic size below 5.0 µm. ED and FPF were determined using the following equations:(1)$$\mathrm{Emitted}\;\mathrm{dose}\;(\mathrm{ED})\%=\frac{Initial\;mass\;in\;capsule-Final\;mass\;remaining\;in\;capsule}{Initail\;mass\;in\;capsule}\;\times \;100\;$$
(2)$$\mathrm{Fine}\;\mathrm{particle}\;\mathrm{fraction}\;(\mathrm{FPF})\%=\frac{Mass\;of\;particles\;on\;stage\;i\;through\;filter}{Total\;mass\;on\;all\;stages}\;\times \;100$$

### 2.5. Cell Culture

Cells were maintained at 37 °C, 5% CO_2_ and grown in a 75 cm^2^ flask in Dulbecco’s Minimum Essential Medium (DMEM) supplemented with 10% *v*/*v* FBS, 1% *w*/*v* L-glutamine. Cells were seeded onto the apical side of the 24-well Transwell^®^ at a density of 5 × 10^5^ cells/cm^−2^ in 100 µL, and 500 µL fresh medium was placed in the basolateral side. For the air-interfaced model, the medium was completely removed on the apical side after 24 h, and the basolateral medium was changed every alternate day.

### 2.6. Transepithelial Electrical Resistance (TEER) Measurement

The Calu-3 cell monolayers were allowed for 12–14 days to differentiate by the air–liquid interface culture method. During this time period, the Calu-3 cell monolayer can be sufficient to ensure adequate tight junction and mucus coverage [29]. TEER of the cell line was measured using a voltohmmeter (EVOM with STX-2 chopstick electrodes, World Precision Instruments, Sarasota, FL, USA). Before resistance measurement, the apical and basolateral sides of the Calu-3 cells were filled and then equilibrated for 30 min. To calculate the TEER value, the resistance of the blank insert was subtracted and multiplied by the surface area (0.33 cm^2^) of the Transwell^®^, as previously described [29].

### 2.7. Cell Surface Staining for Mucus Detection

Alcian blue was used to stain the glycoprotein in the mucus covering the surface of Calu-3 cell monolayers. The Calu-3 cell monolayers were washed twice with 200 μL PBS. The cells were fixed using 4% (*v*/*v*) paraformaldehyde for 15 min and then PBS wash was repeated. Finally, cells were stained with 100 μL Alcian blue for 20 min. The cells were then rinsed with PBS until the rinsate was clear. The filter membrane was cut with a scalpel, mounted on glass slides using rapid mounting medium Entellan™, and sealed. Images were obtained using a CX41 microscope (Olympus, Tokyo, Japan) equipped with a Leica DFC280 digital camera (Leica, Heerbrugg, Germany). Images were analyzed using ImageJ (v1.42q, NIH) with Color Profile (Dimiter Prodanov; Leiden University Medical Center, Leiden, The Netherlands) and Color Inspector 3D v2.0 (Kai Uwe Barthel; Internationale, Medieninformatik, Berlin, Germany) plug-ins. A 3D color space represents the 8-bit red–green–blue (RGB) value of each image. The ratio of blue light (RGB_B_ ratio) was calculated using the following equation:(3)$${\mathrm{RGB}}_{B}\;\mathrm{ratio}=\frac{{\mathrm{RGB}}_{B}}{{\mathrm{RGB}}_{R}+{\mathrm{RGB}}_{B}+{\mathrm{RGB}}_{G}}$$

### 2.8. In Vitro Drug Diffusion Study Using the Calu-3 Cell

The in vitro diffusion of SDL microparticles was evaluated using a Franz diffusion cell system (FCDS-900C, Labfine, Seoul, Korea) to mimic the air–liquid interface of the lung. A schematic diagram of the in vitro drug diffusion study is shown in Figure 1. Briefly, the reservoir of the Franz cell was filled with 10 mL of HBSS and maintained at 37 ± 0.5 °C in a water bath. The capacity of the receptor chamber was 12 mL and the inner diameter was 12 mm. A Transwell^®^ insert with 0.4 μm pore size and 0.33 cm^2^ area available for diffusion was used as the model membrane. A twin stage impinger (TSI) (Copley, Nottingham, UK) was used to deposit the SDL microparticles on Calu-3 cultured Transwell^®^ membranes, which consisted of two stages, and the particles that reached stage two based on the aerodynamic diameter were ≤6.4 μm, when operated at 60 L/min. The airflow was actuated using a vacuum pump at a controlled flow rate of 60 L/min for 5 s (TISCH Environmental, Cleves, OH, USA). Prior to testing, the TSI was connected to a flow meter (DFM 2000, COPLEY Scientific, Colwick, UK) to check the flow rate, and the 4 kpa pressure drop over the inhaler was checked using a critical flow controller (COPLEY Scientific, UK). The SDL microparticles (20 ± 0.2 mg) were weighed into a size 3 HPMC capsule, which was then inserted into an RS01 DPI device. After actuation, the TSI was disassembled and the Transwell^®^ was removed. The particles on the outer surface of the Transwell^®^ were wiped with ethanol-soaked tissue, dried for 15 min at room temperature, and then the Transwell^®^ was transferred to a Franz cell. Samples (200 μL) were collected from the reservoir at predetermined time intervals and replaced with fresh buffer. Total drug amount on the Transwell^®^ was measured as the sum of tiotropium dissolved in the reservoir at 240 min plus remaining on the Transwell^®^ membrane, which was determined by washing the Transwell^®^ membrane with 5 mL of mobile phase at the end of the experiment. The tiotropium was analyzed using validated HPLC methods, as described previously.

### 2.9. In Vivo Study Using Ovalbumin-Induced Asthma Rat Model and Treatment

Sprague-Dawley (SD) rats (8 weeks old) were randomly divided into five groups (*n* = 9). The negative control group was sensitized with 1 mL PBS and challenged with 200 µL PBS. The other groups were sensitized intraperitoneally (I.P.) with 1 mg ovalbumin (OVA) adsorbed on 20 mg Al(OH)_3_ gelatinous in 1 mL PBS on days 0, 7, and 14 and then challenged with 1.1% OVA (200 µL PBS) by intratracheal instillation (ITI) on day 21 [30]. From day 23 to 25, OVA-sensitized and challenged rats were treated with 0.27 mg raw TB microparticles and 1 mg SDL 1 and SDL 4 microparticles with the DP-4 inssuflator™ (Penn-Century Inc., Wyndmoor, PA, USA). The rats were sacrificed on day 27, and the left lung of each rat was obtained for histological analysis.

#### 2.9.1. Measurement of Bronchial Responsiveness

In vivo airway responsiveness to methacholine was measured 24 h after the last treatment using double-chamber plethysmography (DCP, EMKA TECHNOLOGIES, Paris, France). Rat-specific airway resistance (S_Raw_) and enhanced pause (P_enh_) were measured for airway hyper-responsiveness and lung function. The baseline values were recorded while breathing clean air for the first 5 min, treated for 3 min with aerosolized methacholine (25 mg/mL) generated by an ultrasonic nebulizer, and then the breathing patterns for 5 min were recorded and we measured the S_Raw_ and P_enh_ [31]_._

#### 2.9.2. Histology of Lungs

The left lungs of the rats were harvested, transferred into 4% formaldehyde solution for 24 h, and subsequently transferred into PBS. After gradient alcohol dehydration of the left lungs and dewaxing with xylene, they were embedded in paraffin. Finally, slice sections of 4 μm thickness were obtained. The lung sections were stained with hematoxylin and eosin (H&E) and periodic acid–Schiff (PAS).

To evaluate the alveolar structure and distribution of goblet cells, two indicators (goblet cell hyperplasia and airway wall thickness) were examined. To determine goblet cell hyperplasia, quantification was performed using a modified five-point scoring system (grades 0–4) from Padrid et al. [32]. Airway wall thickness was analyzed using the ImageJ software (National Institutes of Health, Bethesda, MD, USA). The thicknesses of 10 airways were measured [33].

## 3. Results and Discussion

### 3.1. Physical Characterization

#### 3.1.1. Particle Morphology

The SEM images of the TB and SDL microparticles are shown in Figure 2. TB had an irregular, aggregated morphology. SDL 1 had an irregular and angular morphology and SDL 2, 3, and 4 had irregularly folded and corrugated morphologies. SDL 1 contained particles that were larger than those observed in SDL 2, 3, and 4, and SDL 1 showed a greater variety of size distributions. This is in agreement with the data obtained using laser diffraction (Table 2). Prior to the drying step, the surface tension of the feedstock was reduced because of the surfactant properties of NaGc. This reduces the size of the droplets produced during atomization and forms smaller particles that favor pulmonary delivery [34].

During the drying phase, water present in the droplets evaporates, and the smaller the size of the droplets, the faster the saturation and crystallization of L-leucine. Therefore, L-leucine crystallization on the smaller droplet surfaces of the SDL 2, 3, and 4 formulations containing NaGc prevented the escape of water vapor and consequently resulted in the formation of corrugated particles [35]. Corrugated particles reduce the contact area between the particles and significantly reduce the van der Waals and electrostatic forces [36]. In addition, the hydrophobic property of L-leucine present on the surfaces of the particles reduces water sorption and the formation of capillary bridges between the particles [37]. As a result, corrugated particles in SDL 2, 3, and 4 formulations are expected to show better aerosol performance than the SDL 1 formulation [38].

#### 3.1.2. PSD

Table 2 presents the particle size and size distribution of the SDL formulations. The data suggest that SDL 2, 3, and 4 had similar particle size distributions, with a median volume diameter (D_v_50) of 2–3 µm. SDL 1 was larger than SDL 2, 3, and 4. However, all the SDL formulations had to be small enough, preferably 1–5 µm in aerodynamic diameter, for lung delivery [1,2,39,40].

#### 3.1.3. XRD

As shown in Figure 3, the XRD diffractograms of SDL formations, raw tiotropium, leucine, sodium glycocholate, and the physical mixture were investigated to characterize the solid state. The diffractograms of raw tiotropium showed sharp diffraction peaks at 12.0, 15.0, 15.7, 20.2, 21.7, 23.8, and 25.4 as 2-theta, indicating high crystallinity. However, the diffractograms of the SDL microparticles showed peaks at 6.1, 19.3, and 32.6 as 2-theta from leucine [41], and there were no specific peaks of tiotropium due to the existence of an amorphous solid state. Furthermore, in SDL 1 microparticles, the intensity of the L-leucine peak was higher than that of other SDL microparticles, and additional peaks were observed at 11.8, 24.3, and 30.8. This is because the ratio of L-leucine in the SDL 1 microparticles was the highest, and it is possible that the crystal structure of L-leucine was weakened by the NaGc in the SDL 2, 3, and 4 microparticles. This was further confirmed by the DSC results.

#### 3.1.4. DSC

As shown in Figure 4, DSC thermograms of raw tiotropium, NaGc, leucine, the physical mixture, and SDL microparticles were investigated to characterize their thermal behavior. The tiotropium and leucine showed endothermic peaks at 232 °C and 320 °C, respectively, which are the melting temperatures (T_m_), indicating that a highly crystalline solid state was achieved. However, the NaGc showed a crystallization temperature (T_c_) of 210 °C and a T_m_ at 243 °C, so it is a partially crystalline solid state. In addition, the T_m_ of the physical mixture shifted to lower temperatures owing to the interaction of the three components of the physical mixture. SDL microparticles composed of tiotropium, L-leucine, and NaGc showed a broad endotherm peak from leucine, and the endotherm peak of tiotropium was not observed, indicating that the crystalline solid state of tiotropium was transformed into an amorphous form. In addition, the higher the ratio of NaGc among SDL microparticles, the more the T_m_ shifted to lower temperatures because NaGc loosened the formation of the crystal structure of L-leucine.

#### 3.1.5. FT-IR Spectroscopy

As shown in Figure 5, the FT-IR spectra of raw tiotropium, NaGc, leucine, the physical mixture, and SDL microparticles were investigated to characterize their functional groups. The region around 2800–3000 cm^−1^ corresponds to carboxylic O-H stretching and -CH3 stretching in leucine [31]. In addition, leucine showed peaks at 1580 cm^−1^, 1509 cm^−1^, and 1403 cm^−1^, attributed to RNH2 bending, CH2 bending, and CH3 bending, respectively. The carbonyl C=O stretching in tiotropium exhibited distinct absorption at 1730 cm^−1^. In addition, it did not change in the SDL microparticles, confirming the absence of chemical changes. Therefore, there were no specific peaks to confirm the chemical changes in the FTIR spectra of the SDL microparticles.

### 3.2. Andersen Cascade Impactor

The aerosol performance of the SDL microparticles was evaluated using an Andersen cascade impactor and a Handihaler^®^ DPI device. Figure 6 shows the %deposition of tiotropium in each ACI stage (−1–6 stage), and the values of FPF, ED, MMAD, and GSD are shown in Table 3. SDL 2, SDL 3, and SDL 4 showed higher FPF values than SDL 1, indicating that the smaller particle sizes and more corrugated morphology resulted in better aerosol performance. The %FPF values were 59.75 ± 2.54%, 68.00 ± 6.51%, 69.12 ± 6.82% and 62.99 ± 6.87% for SDL 1, SDL 2, SDL 3, and SD4, respectively. The MMAD values for SDL 1, SDL 2, SDL 3, and SDL 4 were 2.27 ± 0.31, 1.90 ± 0.18, 1.93 ± 0.05, and 1.88 ± 0.18, respectively. The GSD values for SDL 1, SDL 2, SDL 3, and SDL 4 were 1.41 ± 0.11, 2.15 ± 0.60, 3.32 ± 1.02, and 2.01 ± 0.47, respectively. Particles with corrugated surfaces showed better aerosol performance than smooth-surfaced particles because of the reduced interparticulate van der Waals forces.

### 3.3. TEER Measurements

Monolayer resistance measurements indicated that Calu-3 cells generated a measurable TEER value from day 5 when seeded at 5 × 10^5^ cells/cm^2^ density on the Transwell inserts. There was a significant difference in membrane integrity between day 5 (98.78 ± 34.26) and all other days (575.89 ± 101.21, 529.03 ± 61.81, 452.32 ± 47.50, and 451.26 ± 144.22 on days 7, 10, 14, and 21, respectively). No significant difference was observed in the TEER values from days 5 to 21, indicating that the resistance values reached a plateau (Figure 7). All the established Calu-3 cell models cultured by the ALI method exhibited TEER > 400 Ω·cm^2^_,_ confirming that they formed a tight epithelium. ALI exhibited lower TEER values (450–575 Ω·cm^2^) than the liquid–liquid interface (LLI) TEER values (400–1700 Ω·cm^2^), which is consistent with the findings of others [42,43,44,45]. This could be due to the homogenous, simple epithelium without multilayered regions.

### 3.4. Mucus Detection

Alcian blue differentially detects cellular acidic mucosal changes through the recognition of sialylation (deep blue staining) mucus [46]. A distinct film of acidic mucosa was observed on the apical surface of the Calu-3 epithelium. In general, mucus secretion increased with culture time, as confirmed by calculating the RGB_B_ ratio. Representative microscopic images and a plot of the RGB_B_ ratio with respect to time are shown in Figure 8. Moreover, it was confirmed that the mucus was homogeneously dispersed by increasing the culture time; in addition, the blue area of the RGB color space became smaller and the blue color became deeper. 

### 3.5. In Vitro Drug Permeation Studies Using Calu-3 Cells

The percentages of transferred tiotropium and the P_app_ values are shown in Figure 9. The equation for the P_app_ value is as follows:(4)$$P_{\mathrm{app}}=\frac{\left(\frac{dQ}{dt}\right)}{{\mathrm{AC}}_{0}}$$
where *d*Q/*d*t is the transport rate, A is the surface area of the Transwell membrane (0.33 cm^2^), and C_0_ is the initial concentration of tiotropium on the apical side. For C_0_ calculations, the apical side liquid volume (mucus volume) of the Calu-3 air–liquid interface model was assumed to be approximately 3.44 µL after 11–13 days in culture [26].

The drug percentages of transfer after 240 min of tiotropium for PM, SDL 1, SDL 2, SDL 3, and SDL 4 microparticles were 11.8 ± 4.1, 20.7 ± 6.3, 23.9 ± 4.2, 24.5 ± 1.2, and 36.6 ± 6.9, respectively. The P_app_ values (×10^−7^) were 0.73 ± 0.39 and 1.33 ± 0.56, 1.85 ± 0.44, 0.178 ± 0.10, 2.35 ± 0.7, and 4.13 ± 1.31, respectively. The permeability of tiotropium was significantly increased in SDL 4 microparticles, as the P_app_ value of SDL 4 microparticles was approximately six times faster than that of raw tiotropium. These results suggest that NaGc enhances absorption. The enhancing mechanism of bile salts has not been fully elucidated; however, some plausible mechanisms that may be associated with enhancing the absorption effect are as follows: (1) alteration of mucus layer; (2) protection against enzymatic degradation; (3) increased paracellular absorption due to the opening of tight junctions between Calu-3 cells [47,48]. 

In addition, previous reports by Morimoto et al. concluded that sodium glycocholate may be a safe and useful absorption enhancer for pulmonary drug delivery [49].

### 3.6. In Vivo Effects on Lung Function and Histological Changes in Asthmatic Rats

The airway response to methacholine (25 mg/mL) exposure was measured using double chamber plethysmography (DCP). The P_enh_ values were significantly higher in the PC group than in the NC and SDL 4 groups. However, there were no significant differences between the treatment groups (Figure 10B). In addition, Mch-induced airway resistance values were significantly higher than those in the NC and treatment groups (Figure 10C). This indicates that asthma was well induced by the OVA-induced asthma rat model protocol, and tiotropium, SDL 1, and SDL 4 microparticles significantly reduced airway resistance. H&E (Figure 10D) and PAS staining (Figure 10E) were performed to investigate the effects of tiotropium, SDL 1, and SDL 4 microparticles on the lung tissue of OVA-induced rats. Airway wall thickness was significantly increased in the PC group compared to that in the NC, SDL 1, and SDL 4 groups. However, there was no significant difference between the SDL 1 and SDL 4 groups. Goblet cells secreting mucus from the lungs were quantified by PAS staining. There was a significant increase in the asthma rat model compared to that in the control group (Figure 10G). The PAS scores of the treatment groups decreased, but the difference was not statistically significant. Consequently, in the PC group, P_enh_ and airway resistance increased, and histological changes were observed using DCP and lung tissue staining. In addition, tiotropium, SDL 1, and SDL 4 microparticle-treated groups showed decreased P_enh_ and airway resistance, and decreased airway wall thickness and PAS score, but no significant difference was observed among treatment groups. The time and duration of treatment or the concentration of tiotropium may need to be considered.

## 4. Conclusions

In this study, tiotropium microparticles containing L-leucine with or without NaGc were successfully prepared by spray drying. Physicochemical properties, aerosol performance, Calu-3 cell permeability, lung function, airway responsiveness, and histological changes were investigated. SDL microparticles containing NaGc (SDL 2, 3, and 4) had a size and morphology suitable for inhalation and showed improved deposition in the deep lung region. In addition, the Calu-3 cell permeability study confirmed that the P_app_ values increased with increasing NaGc concentration. Our results demonstrated that the formulation containing NaGc presented better aerodynamic performance and permeability than the formulation without NaGc. In addition, the administration of SDL 4 microparticles to OVA-induced asthmatic rats decreased the P_enh_ and airway hyperresponsiveness to NC group levels. In addition, the airway wall thickness and PAS score decreased in the treatment groups, but there was no significant difference depending on the concentration of NaGc. In conclusion, we provided some insights into possible permeation enhancement strategies using NaGc for pulmonary delivery and the potential for increased pulmonary delivery efficiency. Permeation-enhanced tiotropium delivery via inhalation could result in higher bioavailability and fewer side effects.

## Figures and Tables

**Figure 1 pharmaceutics-14-01409-f001:**
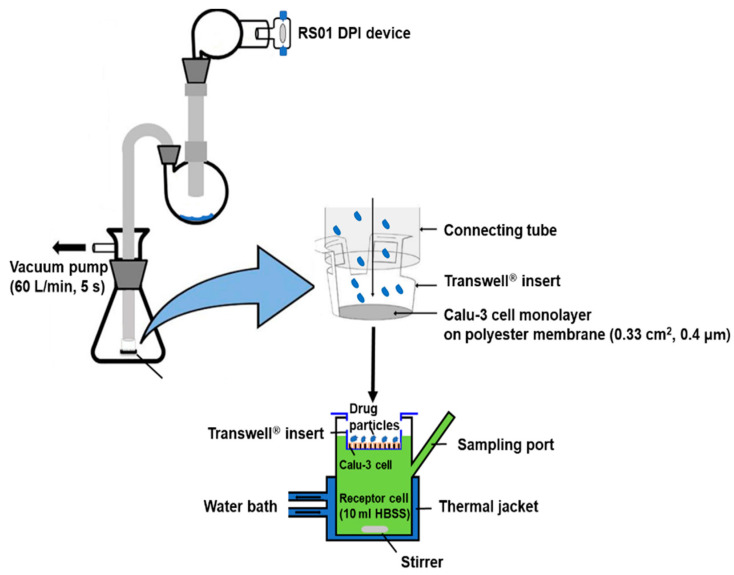
Schematic diagram of the Franz cell setup for in vitro drug diffusion study.

**Figure 2 pharmaceutics-14-01409-f002:**
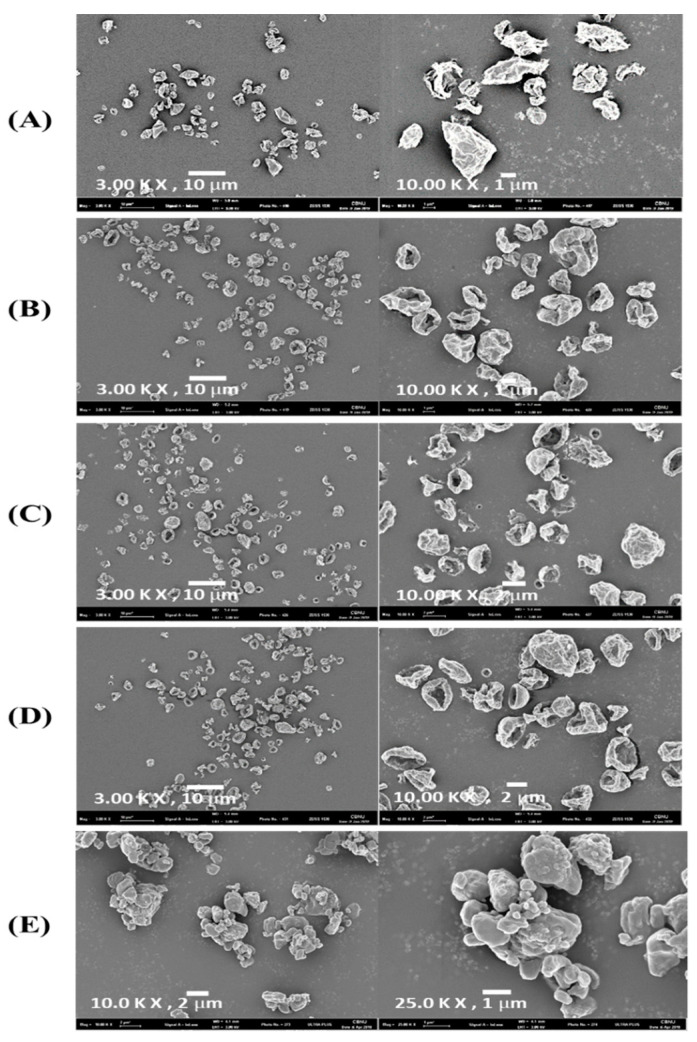
Scanning electron microscopy images of (**A**) SDL 1, (**B**) SDL 2, (**C**) SDL 3, (**D**) SDL 4, and (**E**) raw tiotropium bromide monohydrate at magnifications of 3.0, 10.0, or 25.0 K.

**Figure 3 pharmaceutics-14-01409-f003:**
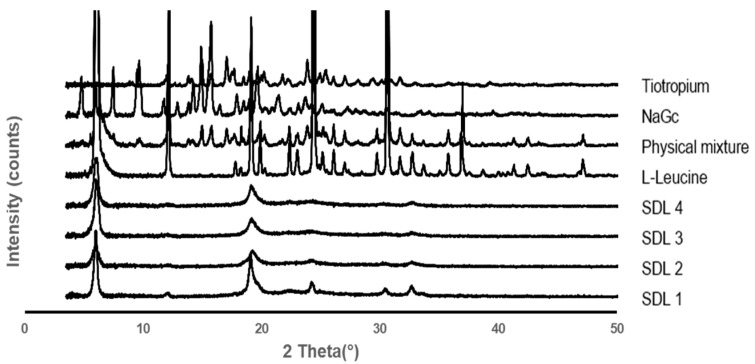
XRD diffractograms of SDL formulations, raw tiotropium, L-leucine, sodium glycocholate, and physical mixture.

**Figure 4 pharmaceutics-14-01409-f004:**
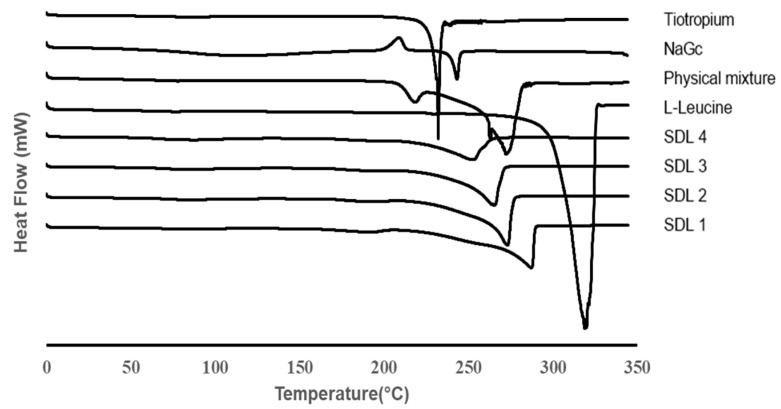
DSC thermograms of SDL formulations, raw tiotropium, l-leucine, sodium glycocholate, and physical mixture.

**Figure 5 pharmaceutics-14-01409-f005:**
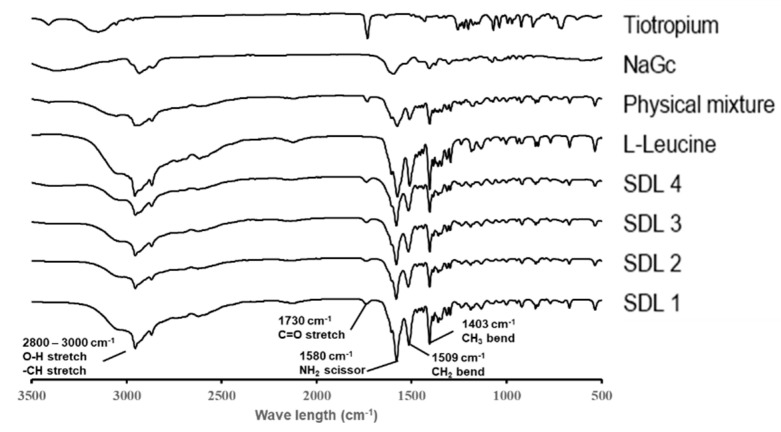
FT−IR spectra of SDL formulations, raw tiotropium, l-leucine, sodium glycocholate, and physical mixture.

**Figure 6 pharmaceutics-14-01409-f006:**
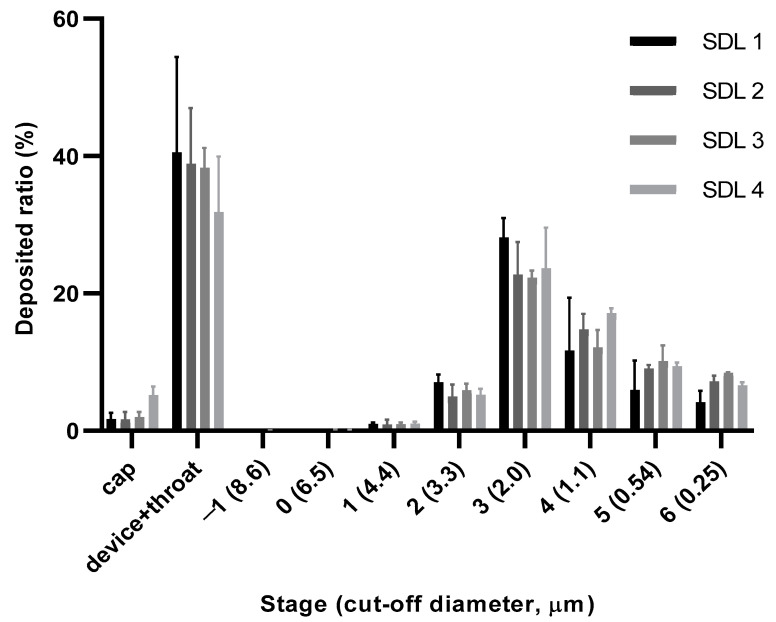
Aerosol dispersion performance as percentage deposited in each stage of Anderson cascade impactor for SDL formulations (Mean ± S.D., *n* = 3).

**Figure 7 pharmaceutics-14-01409-f007:**
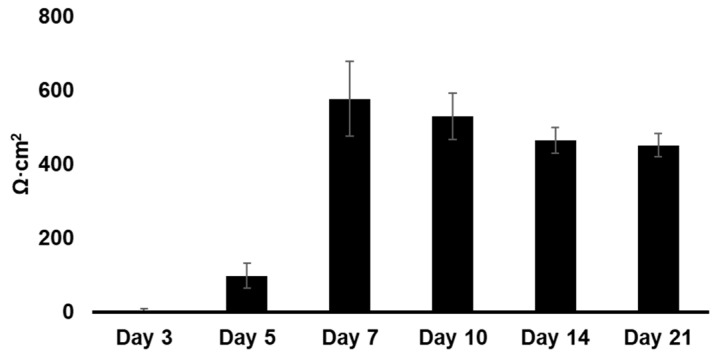
TEER values of Calu-3 cells cultured by the ALI method on Transwell^®^ (Mean ± S.D., *n* = 9).

**Figure 8 pharmaceutics-14-01409-f008:**
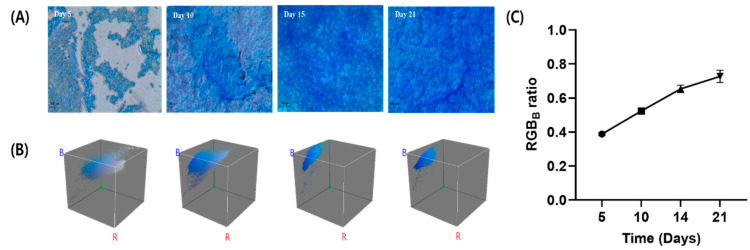
Mucus staining of Calu-3 cells at day 5, 10, 15, and 21. (**A**) Microscope images, (**B**) RGB color space analysis, and (**C**) plot of RGB_B_ ratio as time in culture (Mean ± S.D, *n* = 3).

**Figure 9 pharmaceutics-14-01409-f009:**
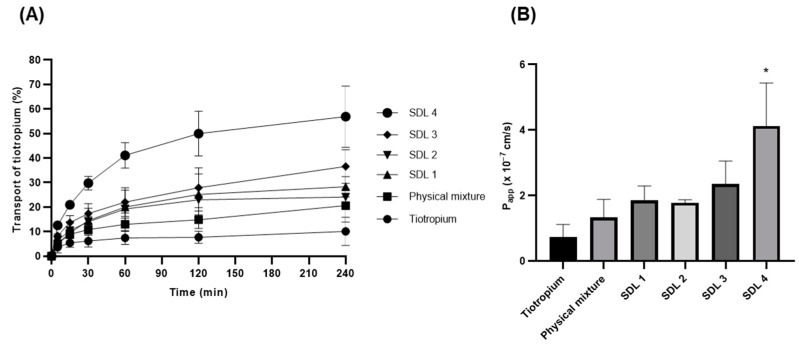
(**A**) Drug transport (apical to basolateral), (**B**) P_app_ values for apical to basolateral of tiotropium, PM, and SDL microparticles (Mean ± S.D., *n* = 4); * significantly different from tiotropium (*p* < 0.005, Kruskal−Wallis test).

**Figure 10 pharmaceutics-14-01409-f010:**
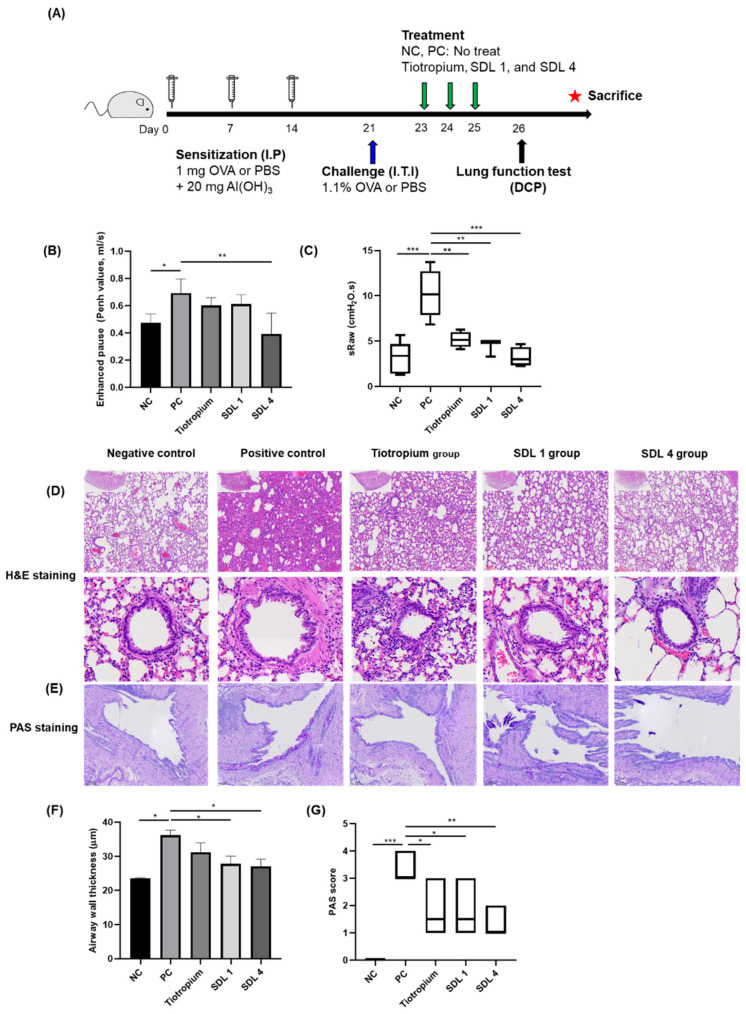
(**A**) Sensitized, challenged with OVA (PC, tiotropium, SDL 1, SDL 4 group) or PBS (NC group) and treatment (tiotropium, SDL1, and SDL 4 group) protocol. (**B**) Enhanced pause values. (**C**) Airway resistance of Mch (25 mg/mL), presented as specific airway resistance (S_raw_) values. (**D**) Representative H&E-stained lung section images for each group. (**E**) Representative PAS-stained lung section images for each group. (**F**) Airway wall thickness (µm). (**G**) PAS score. Statistical analysis was done using ANOVA/Tukey. * *p* < 0.05, ** *p* < 0.01, *** *p* < 0.001, (Mean ± S.D., *n* = 9).

**Table 1 pharmaceutics-14-01409-t001:** Formulation of the spray-dried TB microparticles and physical mixture and content uniformity (*n* = 3, Mean ± S.D).

Formulation	L-Leucine	Tiotropium Bromide	Sodium Glycocholate	Content Uniformity
	(mg)	(mg)	(mg)	(%)
SDL 1	1100	400	-	93.7 ± 0.3
SDL 2	1000	400	100	92.1 ± 4.3
SDL 3	900	400	200	94.5 ± 4.4
SDL 4	700	400	400	95.8 ± 1.5
PM	700	400	400	-

**Table 2 pharmaceutics-14-01409-t002:** Particle size and size distribution of spray-dried formulations.

	SDL 1	SDL 2	SDL 3	SDL 4
D_v_ (10, µm)	2.13	1.04	0.92	0.91
D_v_ (50, µm)	5.36	2.68	2.8	2.55
D_v_ (90, µm)	10.5	4.39	4.4	4.4
Span	1.57	1.25	1.24	1.37

**Table 3 pharmaceutics-14-01409-t003:** Aerosol performance characteristics of tiotropium microparticles including ED, FPF, MMAD, and GSD (Mean ± S.D., *n* = 3).

Formulation	ED%	FPF% < 4.4 µm	MMAD	GSD
SDL 1	98.62 ± 0.68	59.75 ± 2.54	2.27 ± 0.31	1.41 ± 0.11
SDL 2	96.86 ± 0.89	68.00 ± 6.51	1.90 ± 0.18	2.15 ± 0.60
SDL 3	98.04 ± 0.75	69.12 ± 6.82	1.93 ± 0.05	3.32 ± 1.02
SDL 4	94.81 ± 1.25	62.99 ± 6.87	1.88 ± 0.18	2.01 ± 0.47

## Data Availability

Not applicable.

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
