# Peer review of "Preparation and Evaluation of Mucus-Penetrating Inhalable Microparticles of Tiotropium Bromide Containing Sodium Glycocholate"

_pharmaceutics, 2022, doi:10.3390/pharmaceutics14071409_

Round 1

Reviewer 1 Report

This research article shows an extensive characterization of a novel formulation for pulmonary delivery of tiotropium bromide containing permeation enhancers to improve in vivo performance. The formulation is deeply characterized and overall scientific quality is good, however some aspects could be improved:

- English needs to be reviewed, especially in the results and discussion session. Some grammar corrections are needed (e.g. verb conjugation).

- Format also needs to be checked: sometimes a different font is used (lines 309-319, 323-322, etc.), or a different space between lines (section 3.1.5), superscripts where not needed (lines 369-370), extra punctuation/points missing, etc.

- One major concern when reading this paper was the lack of in vitro drug characterization. The HPLC methods mentioned to determine each particle fraction in the cascade impactor studies are not described at all (not even a reference to another paper is given). This needs to be described, as the readers do not know even what was being measured (I guess it was the tiotropium). It would also be interesting to do some in vitro loading tests to check if the amount of drug loaded inside the particles is homogeneous, for example in the different particle fractions.

- Line 246: particles were washed and dried, with what? Does this affect the drug loaded? When using a solvent/water/PBS, early release of the drug can occur.

- Section 3.1.4: change 'shift to the left' to 'shift to lower temperature'

- Figure 8 is missing

- In figure 11 all graphs have a very small font size, I would recommend to organize the figure so they are bigger

Author Response

Thank you for your thorough review and salient observations of this manuscript and for the comments and suggestions, which help to improve the quality of this manuscript

Reviewer 2 Report

In general this is an interesting and worthy publications study. I just have one major remark:

 If Authors use ANOVA then sample size cannot be smaller than the number of groups and yet in this case there are snaple sizes 3 and 4 with numbers of groups exceeding these sizes. Therefore, Authors either will resign from from the conclusion about significance of differences between measureds statistics or use non-paramteric tests.

As I do not see all the sample sizes clearly described with each analysis I kindly ask Authors to provide this information with every figure, where these analyses are shown.

Author Response

Thank you for your thorough review and salient observations of this manuscript and for the comments and suggestions, which help to improve the quality of this manuscript.

Reviewer 3 Report

The work does not demonstrate any major novelties, but only provides the same evidence as what is already known about the ability of permeation enhancers, such as sodium glycocholate, to increase drug permeability  across epithelia.

Different typing characters are used in the manuscript, plus there are different spaces, in the chapters, repeated sentences, missed descriptions etc.

In general the experiments are not comprehensively described in the experimental part and the results are not discussed in relation to a working hypothesis. 

In Particular: 

Line 225: In vitro Drug diffusion study using the Calu-3 cell

It is not clear how the Franz cell has been used. It is reported “ the particles on the outer surface of the Transwell were washed dried and then transferred to a Franz cell. Samples were collected from the reservoir.” Reading this description it can be understood that the particles have been transferred on a membrane in contact with the 10 mL HBSS contained in the reservoir in order to evaluate drug diffusion. However, the description overlooks many important points: what type of membrane has been used in the Franz cell, how the samples collected from the reservoir have been analyzed and especially how the particles have been washed on the outer surface of the Transwell without completely or partially dissolving their components.

Line 418: Mucus detection

There is no description of the impact of treatment on the acidic mucosal changes, therefore it is not clear why the experiment was done.

Line 69-70: the phrase “Absorption enhancers are excipients contained in formulations that overcome biological barriers” is incorrect because their function is not to pass through biological barriers but to help drugs to diffuse through biological barriers 

Line 255: Challenged with 200 uL of what?

Lines 298-301: The phrases have been repeated

Line 309: Supersaturation needs to be proved as it is different from saturation

Line 330 and 347: Physical mixture is reported without specification of its components and relative amounts

Line 400: What is bosentan?

Line 410 and 417: Figure 8 has not been reported 

Line 427: RGB colour space analysis should be explained

Author Response

(The authors gave the same response as above.)

Reviewer 4 Report

Park et al. reported a pulmonary delivery microparticles loading with tiotropium bromide and sodium glycocholate to enhance mucus penetration ability. Overall, the design and experiment were interesting. However, some results and discussions need to be further analyzed and designed. The reviewer suggests a Major Revision before further consideration of this manuscript.

1. In the introduction section, the necessity of mucus penetration was not clearly explained. The fully introduction of respiratory system, mucus barrier and the pathological conditions of asthma should be added to clarify the basis of this research. In this regard, the authors are invited to check some recently published review (e.g., Acta Biomaterialia 119 (2021) 13–29, Advanced Drug Delivery Reviews 185 (2022) 114309, Advanced Drug Delivery Reviews 124 (2018) 140–149)

2. Although has not been very clearly elaborated, the mucus penetration enhancement mechanisms of sodium glycocholate should also be added in the introduction section.

3. Some figures were not presented properly. The lines, characters and figure caption in all figures should be carefully checked and revised. More importantly, Figure 8 was missing in the manuscript.

4. The purpose of TEER measurements and mucus detection was confusing. Is that the constructed air-liquid interface was employed to evaluate the Papp value shown in Figure 10? However, according to the methods described in section 2.8, some details of the Transwell system was missing, such as the cells monolayer culture time and mucus secretion time. It is hard to understand the connection between Figure 8, 9 and 10.

5. An asthma model was employed to evaluate the efficiency of SDL. However, the data of mucus secretion in such a model and the in vivo mucus penetration enhancement of SDL was not provided, which is critical for this mucus penetration microparticles.

6.  The formats of reference should be carefully checked. Further, the format of the main text should also be double-checked.

Author Response

(The authors gave the same response as above.)

Round 2

Reviewer 1 Report

Most of my concerns in the first review round have been addressed. However I still think some English should be corrected (for example, lines 210-212, the article is missing sometimes and verb conjugations are still not correct all the time). The article is perfectly understandable but I believe a thorough look at this would improve the quality very much.

I would also recommend to include in the paper the content uniformity of drug (I guess it is expressed in %?) inside the particles and how it was measured, as it is available data that you shared in your first reply to the review.

Author Response

(The authors gave the same response as above.)

Reviewer 2 Report

Dear Authors

I believe, I specifically pointed out to the non-parametric tests. t-Student is a parametric test.

Author Response

(The authors gave the same response as above.)

Reviewer 3 Report

Line 83 lung not ling
Line 88-89 and 97-100
Check English.
Line 258
The description of the in vitro drug diffusion using Calu-3 cells is not clear to the reader. A Figure
or a schematic drawing describing the whole apparatus must be included in the manuscript.
In Particular:
It is reported that “A 263 Transwell® insert with 0.4 μm pore size and 0.33 cm2 area available for
diffusion 264 was used as the model membrane”
Were the cells on the Transwell insert?
Have the collected samples from the reservoir been analyzed for drug content?
How the drug dissolution rate was evaluated?
How the drug diffusion rate was evaluated?
At what time points was the total drug deposition on the Transwell determined by the sum of
the amount of drug remaining in the membrane?
If the Transwell was washed at any time point there was no drug left in the Transwell to add to
other time points.
Why was the drug remaining in the membrane dissolved in the receptor phase?
Line 341
Supersaturation does not always take place in drying solutions as it is a transient condition that
must be demonstrated if it is mentioned.
Line 446
The mucus detection experiments still seem disconnected from the other experiments. They
should also be performed with PM and SDL to see if the presence of the penetration enhancer
modifies the mucus layer. This aspect is of great importance in determining the efficacy and
tolerability of inhalable microparticles, therefore it should be discussed.
Line 453
The mucus was homogeneously covered? Or dispersed?
Figure 9 C is not legible.

Author Response

(The authors gave the same response as above.)

Reviewer 4 Report

The authors have addressed most of issues I raised. However, the formats of Figures are still far from requirement. Specifically, the thickness of lines should be uniform over through the manuscript. The lines in Figure 4, Figure 5, Figure 8 and Figure 9C are too thin, and the characters in the axis are in unproper color and font. Further, the authors should provide a clear copy without revision trace to be reviewed.  

Author Response

(The authors gave the same response as above.)

Round 3

Reviewer 2 Report

No comments. Thank you.

Author Response

Thank you for your thorough review, which help to improve the quality of this manuscript.

Reviewer 3 Report

Line 69 

‘In this study, we used bile salt as a permeability enhancement. They have been.......’

The sentence should be revised: is salt singular or plural?

Line 253 

20mg, ±  

The comma is necessary?

Line 255-256

‘The particles on the outer surface of the Transwell® were wiped with ethanol soaked tissue, dried for 15 min at room temperature, and then transferred to a Franz cell’

The sentence should be revised it looks like the wiped particles are transferred to the Franz cell

Author Response

(The authors gave the same response as above.)
